# Factors Affecting Population Dynamics of *Helicoverpa zea* (Lepidoptera: Noctuidae) in a Mixed Landscape with Bt Cotton and Peanut

**DOI:** 10.3390/insects14040395

**Published:** 2023-04-19

**Authors:** Izailda Barbosa dos Santos, Silvana V. Paula-Moraes, Julien M. Beuzelin, Daniel A. Hahn, Omaththage P. Perera, Clyde Fraisse

**Affiliations:** 1West Florida Research and Education Center, Department of Entomology and Nematology, University of Florida, Jay, FL 32565, USA; 2Everglades Research and Education Center, Department of Entomology and Nematology, University of Florida, 3200 East Palm Beach Road, Belle Glade, FL 33430, USA; 3Department of Entomology and Nematology, University of Florida, 1881 Natural Area Drive, Gainesville, FL 32611, USA; 4Southern Insect Management Research Unit, USDA-ARS, Stoneville, MS 38776, USA; 5Department of Agricultural and Biological Engineering, University of Florida, 271 Frazier Rogers Hall, Gainesville, FL 32611, USA

**Keywords:** isotopic carbon, cotton belt, migration, corn earworm, cotton bollworm

## Abstract

**Simple Summary:**

*Helicoverpa zea* is a polyphagous pest of multiple cultivated and non-cultivated plants throughout the Americas. Overall, in North America, *H. zea* overwinters in southern states and recolonizes areas in northern states, which have lethal winter temperatures for this species. Studies of *H. zea* ecology in southern states, such as Florida, which are a source of migrants, can provide region-specific information with potential continental-wide implications. Here we determined the occurrence of *H. zea* in commercial fields of Bt cotton and peanut in two regions of the Florida Panhandle, investigated the effect of weather on the moth’s occurrence, and determined hosts used by *H. zea* across seasons from 2017 to 2019. The results indicate a year-round flight of *H. zea* and no difference on catches among traps set on Bt cotton versus peanut fields. The highest and lowest *H. zea* moth catches were from July to September and November to March, respectively. Higher *H. zea* catches were observed in warm and humid conditions. Based on carbon isotopic analysis, we detected larval feeding predominantly on C3 host plants throughout the year, which includes Bt cotton, with additional larvae feeding during the summer on C4 hosts, which includes Bt corn. These results suggest that overwintering and resident populations of *H. zea* in the Florida Panhandle may have increased exposure to Bt crops, increasing the risk for the evolution of resistance.

**Abstract:**

In North America, weather and host-plant abundance drive the population dynamics of the migratory pest *Helicoverpa zea.* The objectives of this study were to (i) estimate monthly abundance of *H. zea* moths in Bt cotton and peanut fields, (ii) document the effects of weather on *H. zea* trap catches, and (iii) determine larval hosts supporting *H. zea* populations from 2017 to 2019. Year-round trapping of *H. zea* moths was conducted in 16 commercial fields in two regions of the Florida Panhandle using delta traps. *H. zea* moth catches were associated with temperature, rainfall, and relative humidity. Larval hosts were determined by isotopic carbon analysis. Our results showed year-round *H. zea* flights in both regions across two years, with the highest and lowest moth catches occurring from July to September and November to March, respectively. There was no difference in catches between traps set on Bt cotton and peanut. In the Santa Rosa/Escambia counties, weather explained 59% of the variance in *H. zea* catches, with significant effects of temperature, relative humidity, and rainfall. In Jackson County, weather explained 38% of *H. zea* catches, with significant effects of temperature and relative humidity. Carbon isotopic data showed that feeding on C3 plants, including Bt cotton, occurred over most of the year, although feeding on C4 hosts, including Bt corn, occurred during the summer months. Hence overwintering and resident populations of *H. zea* in the Florida Panhandle may be continually exposed to Bt crops, increasing the risk for the evolution of resistance.

## 1. Introduction

*Helicoverpa zea* Boddie (Lepidoptera: Noctuidae) is a polyphagous pest of more than 100 hosts across 21 families of plants, including multiple cultivated crops and wild hosts [1,2,3,4,5]. This species has a lower developmental temperature threshold of 12.5 °C, and in North America, winter temperatures, combined with host plant availability and abundance, drive *H. zea* population densities and distributions [6,7,8,9]. In the southern United States, *H. zea* is multivoltine [5,10], infesting multiple hosts throughout the year [11,12]. Stable carbon isotope (δ13C) analysis of moth wings has been used to investigate larval hosts of *H. zea* in the landscape among plants that produce three- (C3) or four- (C4) carbon compounds during the first step of the carbon fixation cycle in the photosynthetic pathway [13,14,15]. Generally, in the southeastern United States, the first generation of *H. zea* mainly uses C3 wild host plants such as crimson clover (*Trifolium incarnatum*) [4], pepperweed (*Lepidium* sp.) [3], and wild toadflax (*Nuttallanthus texanus*) [1,5]. The second generation feeds on corn (*Zea mays* L.), a C4 plant, and subsequent generations switch to late-season C3 crops such as cotton (*Gossypium hirsutum* L.), soybean (*Glycine max* L.), and peanut (*Arachis hypogaea* L.) [14].

Genetically engineered corn and cotton traits expressing *Bacillus thuringiensis* (Bt) toxins have been commercialized in the United States for more than 20 years to manage pest insects, including *H. zea* [16,17]. The first Bt crop traits only expressed a single toxin (i.e., Cry1Ab in corn and Cry1Ac in cotton) [18]. In subsequent traits, other toxins were pyramided to broaden the range of target pests and reduce the risk of resistance evolution [16,19,20]. Currently, multiple Bt corn and cotton traits are commercially available, and they express a combination of Cry1, Cry2, and Vip toxins [20]. In the southeastern United States, corn is cultivated during spring and summer, while cotton is cultivated in the summer and fall. The temporal separation of corn and cotton can create a sink-to-source relationship of *H. zea* feeding following the availability and attractiveness of these crops, potentially selecting genes that confer resistance to Bt toxins [13,14,16]. Due to the long-range migratory capacity of *H. zea* [6,21,22,23,24], the spread of Bt-resistance genes to other regions is a concern. This is especially a concern in North America, where temperature seasonality creates zones that are thermally unsuitable for the year-round survival of *H. zea* resident populations [6,9,25]. In the U.S., *H. zea* can overwinter in southern states and migrates to northern states during the early spring and summer months [6,23]. Hence, population dynamics and pest management throughout the southern states can influence the densities and genetic makeup of the *H. zea* populations that migrate to northern regions.

The Florida Panhandle, located around latitude 30th N parallel, is a potential overwintering site for *H. zea*. The Florida Panhandle agricultural landscape consists of multiple field crops and vegetables that are hosts for *H. zea*, including cotton, peanut, corn, sweet corn, wheat (*Triticum aestivum* L.), and soybean [26]. Due to the presence of natural refuges for this pest [13,14], structured refuges are not required in southern Bt cotton-growing areas [27], and 100% of cotton fields in the Florida Panhandle express Bt toxins. As both corn and cotton are cultivated in the Florida Panhandle, the succession of these Bt hosts might place intense selection pressure on *H. zea* populations, potentially leading to decreased susceptibility to Bt toxins over time [16].

During the period of the present study, concentration-response bioassays with *H. zea* populations from Florida using the Cry1A.105 toxin detected lethal concentrations to kill 50 percent (LC_50_) of exposed larvae below 0.2 μg/cm^3^ in 2017 [28], 0.27 in 2018 [29], and above 10 μg/cm^3^ in 2019 [29]. Because the Florida Panhandle is a source of *H. zea* migrants, ecological studies of the *H. zea* population in this region can provide region-specific information for *H. zea* management with potential continental-wide implications. To better predict *H. zea* population dynamics and succession of host usage in the region, the objectives of this study were to (i) document *H. zea* monthly abundance in Bt cotton and peanut fields, (ii) estimate the effects of weather on *H. zea* trap catches, and (iii) determine larval hosts supporting *H. zea* populations across multiple seasons from 2017 to 2019.

## 2. Materials and Methods

### 2.1. Pheromone Trapping and Insect Collection

*Helicoverpa zea* flight was monitored by trapping in two regions of the Florida Panhandle from June 2017 to June 2019. The first region was represented by eight commercial fields within Jackson County, and the second region by eight fields within Santa Rosa and Escambia counties. Santa Rosa and Escambia were considered one region for this study due to their proximity. These counties were selected due to their high acreage of cultivated Bt cotton and peanut [26]. Four commercial fields averaging 25 hectares of Bt cotton and four fields of peanut were selected in each region, and two delta Pherocon^®^ VI Traps (Trécé Incorporated, Adair, OK, USA) were mounted on the edges of each field on aluminum poles, approximately one meter above the ground to collect *H. zea* moths (Appendix A). In each region, four of the selected fields (two Bt cotton and two peanut fields) were under irrigation and the other four fields were rainfed.

Trap sticky liners were replaced every two weeks, and sex pheromone lures (Hercon Luretape, Hercon Environmental # HC-3132-10) were replaced every four weeks. After replacement, the collected sticky liners were placed inside 32 × 23 cm resealable plastic bags (S. C. Johnson, Racine, WI, USA) and brought back to the laboratory to be stored in a −20 °C chest freezer at the West Florida Research and Education Center (WFREC). Trapped *H. zea* moths were identified using morphological characteristics, such as the black-colored spot on the forewings, a broad dark transverse band distally, and hind wings lighter in color [6].

Pheromone trapping was conducted year-round, and the total number of *H. zea* moths collected in both traps was determined for each field on each sampled date. During sticky liner replacement, information on the region, date, crop, and irrigation system (e.g., irrigated or rainfed) was also recorded. In 2018, three fields in Santa Rosa/Escambia counties and one field in Jackson County had corn cultivated during the spring before peanut and Bt cotton were planted. Preliminary analysis revealed no difference in *H. zea* monthly abundance with or without data of *H. zea* moths trapped in corn fields. Due to the limited number of observations collected from corn fields in the agroecosystems under study, the corn-trapped moths’ data were removed from the dataset in further analysis for consistency.

### 2.2. Carbon Isotopic Signatures of H. zea Moths

A sample of trapped *H. zea* moths was selected and submitted to carbon isotopic analysis to identify larval feeding on C3 or C4 host plants. A total of 95 *H. zea* moths (13, 33, 34, and 15 moths for the spring, summer, fall, and winter, respectively) were selected from Jackson, and 105 (14, 43, 40, 8 moths for the spring, summer, fall, and winter, respectively) were selected from Santa Rosa/Escambia counties. For consistency of data analysis and body symmetry variability [30], only the right forewings were used for analysis. Collected wings were stored in 4 mL glass vials at −20 °C until analysis. The samples were sent to the Stable Isotope Mass Spec Lab, at the University of Florida, and δ13C signatures were determined using a Thermo Delta V Advantage isotope ratio mass spectrometer coupled with a ConFlo IV interface attached to a Thermo IsoLink elemental analyzer. Standard delta notation relative to V-PDB (Vienna Pee Dee Belemnite) was used to express carbon isotopic results. Moths with a δ13C ratio below −20‰ were categorized as having C3 larval hosts and those with a δ13C ratio above −15‰ as having C4 larval hosts. Adults with a δ13C ratio ranging from −20 to −15‰ were considered as coming from larvae that had mixed C3 and C4 hosts [13].

### 2.3. Weather Data

Average daily measurements of air temperature, relative humidity, and accumulated rainfall were obtained for each region from a Florida Automated Weather Network (FAWN) station near the trapping sites. For traps set in Jackson County, weather data were collected from the FAWN station located in Marianna (GPS coordinates: 30.85000, −85.16516). For traps set at Santa Rosa/Escambia, weather variables were collected from the FAWN station located in Jay (GPS coordinates: 30.77516, −87.14015).

### 2.4. Statistical Analysis

All analyses were performed in R (version 4.2.0) and R studio (version 2022.02.3) [31]. In the first model, we tested the effects of the region on moth abundance using a generalized linear mixed model (GLMM) with the region used as the fixed effect and crop, irrigation system, month, and farm as random effects. Due to a zero inflation effect, the model was refitted using the package glmmTBM with a similar model structure and an inflation parameter (zi = 1) [32].

In the second model, we tested whether *H. zea* abundance differed by month in each region using a GLMM. Month was set as the fixed effect, and crops, irrigation system, and farm were set as random effects. Using a third model, we tested the effect of crop (peanut or Bt cotton) and irrigation system (irrigated or rainfed) in *H. zea* moth trap catches for each region using a GLMM. Crop and irrigation systems were set as fixed effects and farm and month were set as random effects. For the second and the third models, type III Wald Chi-Square tests were used to check the significance of fixed effects on *H. zea* moth catches, and means were compared by multiple pairwise comparisons using least-square means at α = 0.05 (package emmeans) and the Sidak correction was used to control family wise error rate [33].

To test if the larval feeding category (C3, C4, and mixed) proportions were different in each region we calculated the proportion (0 to 1) for each feeding category per season (n = 4) and performed a Kruskal–Wallis test, and compared means using the Wilcox test with Bonferroni correction.

The association between weather variables and the abundance of *H. zea* moths caught was analyzed using a generalized linear model (GLM), setting average daily temperature, relative humidity, and accumulated rainfall as predictors and the number of captured moths as the response for each location. The correlation among variables was tested before analysis using the corrplot package. As all correlation values were below 0.70 (r ≤ |0.70|), all variables were used in the model [34].

In all models, negative binomial and Poisson error distributions were tested, but the negative binomial distribution (link = log) fit the data better. Model suitability (dispersion, zero inflation, residual distribution, uniformity, and outliers) was checked using the packages DHARMa and performance [35], and models were refitted when needed as previously described.

## 3. Results

### 3.1. Month Abundance

*H. zea* moths were caught year-round in both regions (i.e., Jackson, and Santa Rosa/Escambia) in the Florida Panhandle. Santa Rosa/Escambia had a higher (2317 moths) total *H. zea* catch than in Jackson (1619 moths) (χ^2^ = 7.88, *p* = 0.005). In both regions, the highest *H. zea* abundance was in late summer, from July to September, and the lowest abundance period was late fall and overwinter, from November to March (Figure 1).

No effect for crop, irrigation system, or crop x irrigation system interaction was observed for trap catches in Jackson County (Table 1). However, a significant effect was observed for the irrigation system (χ^2^ = 4.45, *p* = 0.03) in Santa Rosa/Escambia (Table 1), with higher catches in rainfed fields.

### 3.2. Carbon Isotopic Signatures of H. zea Moths

*H. zea* moths caught in Jackson County had δ13 C values ranging from −31 to −11 (Figure 2a). The highest proportion of C4 larval host feeding occurred during the summer (27%), followed by winter (24%) (Figure 2b). There was a significant difference in larval feeding among C3, C4, and mixed host categories (Kruskal–Wallis χ^2^ = 7.54, df = 2, *p*-value = 0.023). Sixty-two percent of the *H. zea* moths analyzed came from larvae fed on C3 host plants (Figure 2c).

In Santa Rosa/Escambia, δ13 C values ranged from −32 to −10 (Figure 3a). The proportion of C4 larval hosts was 0.21, 0.33, and 0.23 during spring, summer, and fall, respectively (Figure 3b). There was a significant difference in larval feeding among C3, C4, and mixed host categories (Kruskal–Wallis χ^2^ = 7.38, df = 2, *p*-value = 0.025). On average 60% of the evaluated moths developed on C3 hosts (Figure 3c).

### 3.3. Association of Moth Catches and Weather in the Florida Panhandle from 2017 to 2019

Throughout the duration of this trapping study the mean air temperatures in Jackson and Santa Rosa/Escambia counties were 20.6 °C and 19.6 °C, respectively (Figure 4a,b). The relative humidity in Jackson varied from 40 to 98%, with an average of 79% (Figure 4c). In Santa Rosa/Escambia relative humidity ranged from 49 to 100% and averaged 84% (Figure 4d). In both regions, the lowest relative humidity averages were recorded during the spring. Rainfall occurred year-round in both locations, with the highest averages during the summer (Figure 4e,f).

Weather significantly affected *H. zea* moth catch (Table 2). Higher catches occurred during warm and humid seasons (Figure 5 and Figure 6). In Jackson County, weather explained 38% of the variation in trap catch with significant effects of temperature and relative humidity (Table 2, Figure 5). In Santa Rosa/Escambia, weather explained 59% of the variation in *H. zea* moth catch with significant effects of temperature, relative humidity, and rainfall (Table 2, Figure 6).

## 4. Discussion

Here, we report year-round *H. zea* flights within the Florida Panhandle, a region with a mixed landscape of Bt and non-Bt crops. From spring 2017 to spring 2019, higher *H. zea* moth catch numbers were observed in Santa Rosa/Escambia than in Jackson County. Based on isotopic carbon analysis, plants with the C3 physiologies were the main *H. zea* larval hosts throughout the year. However, an increase in larval feeding on C4 plants was observed in insects collected during the summer and winter in Jackson County, and in the summer in Santa Rosa/Escambia counties. Weather influenced *H. zea* trap catches, and higher catches were observed during warm and humid conditions.

In this study we detected a significant difference in *H. zea* trap catches between Jackson and Santa Rosa/Escambia counties. This difference might be in part due to the large acreage of transgenic cotton and corn expressing Bt toxins. During the period of *H. zea* trapping, the acreage of Bt cotton and corn (potentially also expressing Bt toxins) in Jackson was higher (46,874 ha) than in Santa Rosa/Escambia (31,678 ha) [26]. A previous life table study with *H. zea* populations from Florida Panhandle [36] detected lower growth rate, survival, and body weight for *H. zea* feeding on dual gene Bt cotton. In addition, transgenic cotton expressing Vip3Aa caused 100% *H. zea* larval mortality [36], reinforcing the high efficacy of this toxin for *H. zea* management [37] and the potential role of Bt cotton as a sink habitat to susceptible *H. zea* populations in this landscape [38,39]. Area-wide suppression of target pests by Bt crops with consequent reduction of broad-spectrum insecticides use has been reported [40,41,42,43]. In Mississippi, for example, a 20-year trapping study (1986–2005) detected a drastic decrease in *H. zea* populations after the use of Bt transgenic crops [44]. Thus, larger acreage of Bt crops in Jackson, especially traits carrying Vip3Aa, may act as a sink crop, reducing *H. zea* population densities [44,45].

A significant effect was observed for the irrigation system in Santa Rosa/Escambia, with higher catches in rainfed fields. Accordingly, higher *H. zea* larval densities have been detected in rainfed soybeans fields [46], and irrigation has been shown to cause *H. zea* pupal mortality in cotton fields [47]. The impact of irrigation on the immature stages of *H. zea* could explain the effect of irrigation on Santa Rosa/Escambia trap catches. However, this study focused on determining the monthly abundance of *H. zea* moths, which limits us in drawing further conclusions about factors regulating *H. zea* immature survival in rainfed, and irrigated fields.

Carbon stable isotope analysis indicated that C3 plants are the primary host supporting *H. zea* populations in two different test sites within the Florida Panhandle, separated by approximately 200 km. Bt cotton and peanut are the main C3 crops cultivated in Florida Panhandle [26]. No differences in *H. zea* moth catches were detected between traps set on Bt cotton versus peanut crops. Considering the cultivation of Bt cotton in the region and the efficacy of this technology on suppressing *H. zea* populations [36] peanut is expected to support large populations of *H. zea* larvae, playing a role as a source of moths with C3 larval hosts in the Florida Panhandle. Due to *H. zea* flight capacity [23,24], moths trapped in Bt cotton fields could have originated from larvae that fed on peanut or other cultivated and non-cultivated C3 hosts. Interestingly, the peak of *H. zea* abundance detected here corresponds to when cotton is in its reproductive stage, a highly attractive phenological stage for *H. zea*. Thus, cotton fields in Florida Panhandle should be monitored as a proactive approach to detect early escapes of this Bt technology and to determine the role of cotton as a source of C3 feed moths.

A mixed feeding on both C3 and C4 plants was observed in all seasons. As *H. zea* is a polyphagous pest, larval movement between C3 and C4 crops and weeds could explain the occurrence of adults with mixed feeding [48,49]. Besides, an increase in C4 feeding during the summer was documented in both regions, and it may be attributed to corn cultivation at the end of the spring, and the beginning of the summer in the Florida Panhandle [50]. Although corn is less than 5% of the total acreage cultivated in the region, this crop can be considered the prevalent plant with C4 photosynthetic pathway in both landscapes during the crop season. Furthermore, the effect of corn on the increase of the local population of *H. zea* during the crop season has been documented in a landscape with corn and Bt cotton [51]. Thus, corn may serve as an early-season host for *H. zea*, which later feeds on late-season Bt cotton fields. Bt corn and Bt cotton expressing similar toxins exert continuous selection pressure for Bt resistance on multiple generations of *H. zea* [52]. This concern is particularly relevant for the Vip3A toxin, and removal of this trait from Bt corn has been advocated as a critical insect resistance management (IRM) strategy for *H. zea* [16].

An increase in C4 larval feeding in Jackson County was detected during the winter. Previous studies with the carbon isotopic signature of *H. zea* hypothesized C4 larval feeding when corn was not present in the landscape as an indication of the influx of migrant moths [13,14]. Similarly, *H. zea* migratory populations might be arriving in the region under study during the fallow season. Florida is one of the largest producers of sweet corn in the U.S. ($137 million in 2020) [53], and this production is concentrated in the southern part of the state, during the fall and winter seasons from October to March [26,54,55]. Besides sweet corn, field corn cultivated for silage and grain in the Southern and Central parts of Florida can also host *H. zea* [26,56,57]. Jackson is separated by 200 km from Santa Rosa/Escambia counties, and possible differences in wind direction [24,58] could explain the differential influx of *H. zea* migratory populations into the Florida Panhandle. Although we expect C4 migrants to originate from field and sweet corn from South Florida, more studies are needed to determine the contribution of crop hosts and wind patterns on *H. zea* migratory paths in North America.

Climate plays a role in *H. zea* population dynamics at regional and continental scales [22,24]. In this study, weather explained 38% and 59% of *H. zea* moth catch in the Jackson and Escambia/Santa Rosa counties, respectively. In Jackson County, temperature and relative humidity influenced *H. zea* catch, while temperature, relative humidity, and rainfall influenced catch in Santa Rosa/Escambia counties. As poikilothermic organisms, faster development, higher population densities, and higher flight activities could explain higher moth capture under higher temperatures [59]. A strong positive relationship between *H. zea* moth catch and precipitation has been hypothesized to be an indirect effect of rainfall on host abundance and suitability [60,61]. Likewise, an increase in humidity often increases moth catch in pheromone traps. For example, the interaction between sex pheromone and humidity receptors has been shown to play a role in the European corn borer, *Ostrinia nubilalis* (Hubner), sex pheromone perception [62].

The lowest *H. zea* moth abundance occurred from November to March. The low and sporadic numbers of *H. zea* caught during the winter corroborate previous studies in Mississippi [44]. Although *H. zea* moths were caught in low numbers throughout the winter in both regions of study, the year-round presence of *H. zea* moths in the landscape is a strong indication that the Florida Panhandle may be an *H. zea* overwintering site. This result is relevant to *H. zea* IRM programs. Multiple *H. zea* generations in this region [10] could result in resistant individuals selected in both Bt corn and cotton during a crop season, which could then overwinter and be a source of pest infestations in the following year. In addition, *H. zea* populations could undergo multiple episodes of selection for resistance alleles that could later be spread northward on the American continent. Hence, we expected that adults collected from November through March are probably (i) moths coming from larvae or pupae that had an extended developmental time, or (ii) migratory moths from southern Florida and possibly the Caribbean.

We conclude that *H. zea* moths occur all year round in the landscape of the Florida Panhandle and the peak of flight occurs from July to September. Pheromone traps catch increases in higher temperature and humidity. Additionally, there is a predominance of *H. zea* moths derived from C3 hosts. During the summer, a substantial number of *H. zea* moths derive from C4 hosts, possibly due to the cultivation of corn, a preferred host of *H. zea*. Population dynamics, regional host occurrence (both wild and cultivated), weather, and *H. zea* migration dynamics within Florida may result in differential selection pressure for Bt and insecticide resistance.

## Figures and Tables

**Figure 1 insects-14-00395-f001:**
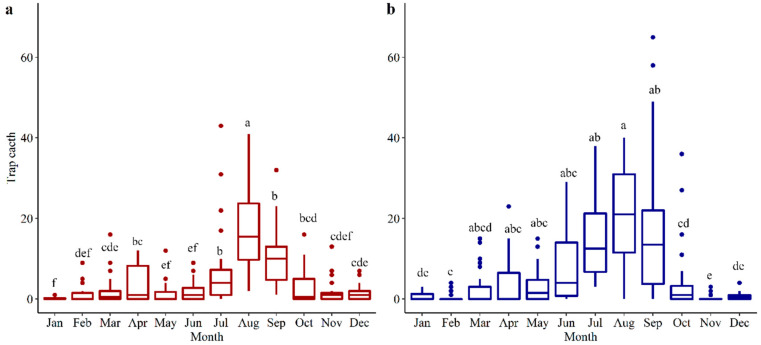
Number of *H. zea* moths trapped per month (**a**) Jackson and (**b**) Santa Rosa/Escambia counties. Boxplots with the same letters are not significantly different from each other (*p* > 0.05). Boxplots indicate the range of data dispersion (first and third quartiles and extreme values), median (solid horizontal line), and outliers (dots falling outside lines).

**Figure 2 insects-14-00395-f002:**
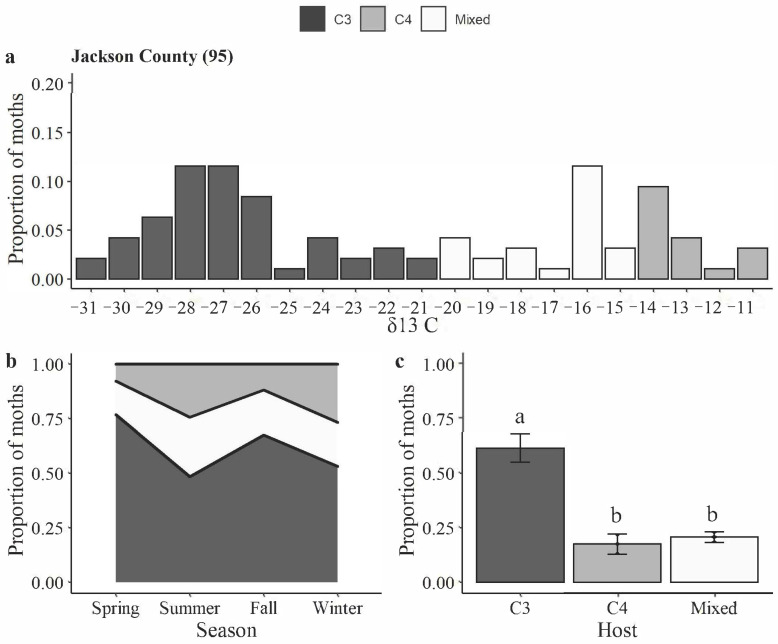
Carbon isotopic analysis of *H. zea* moths trapped in commercial fields in Jackson County: (**a**) distribution of δ^13^ C values from *H. zea* wings (*y*-axis indicates the proportion of samples with a given δ^13^C value); (**b**) proportion of insects developed on C3, C4, or mixed (both C3 and C4) host plants over the seasons; and (**c**) proportion of insects developed on C3, C4, or mixed host plants. Bars with the same letters are not significantly different (*p* > 0.05, Wilcox test).

**Figure 3 insects-14-00395-f003:**
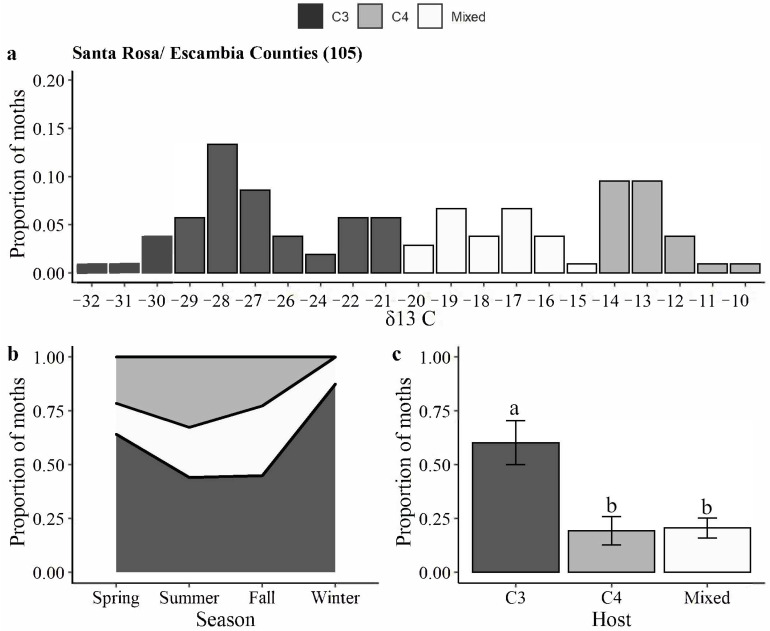
Carbon isotopic analysis of *H. zea* moths trapped in commercial fields in Santa Rosa/Escambia counties: (**a**) distribution of δ^13^ C values from *H. zea* wings (*y*-axis indicates the proportion of samples with a given δ^13^C value); (**b**) proportion of insects developed on C3, C4, or mixed (both C3 and C4) host plants over the seasons; and (**c**) proportion of insects developed on C3, C4, or mixed host plants. Bars with the same letters are not significantly different (*p* > 0.05, Wilcox test).

**Figure 4 insects-14-00395-f004:**
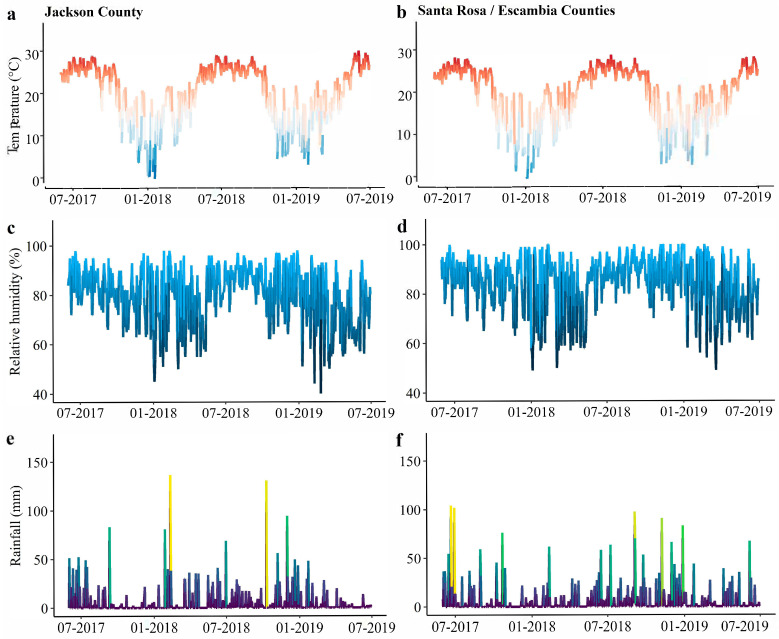
(**a**) Average daily temperature, (**c**) relative humidity, (**e**) and daily accumulated rainfall, from 2017 to 2019 in Marianna, Jackson County. (**b**) Average daily temperature, (**d**) relative humidity, and (**f**) daily accumulated rainfall from 2017 to 2019 in Jay, Santa Rosa County.

**Figure 5 insects-14-00395-f005:**
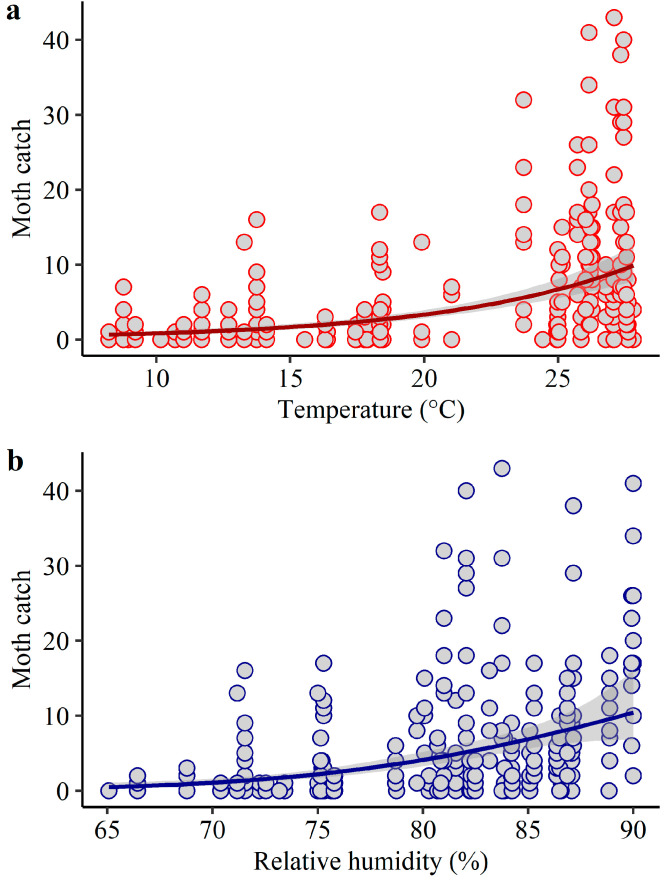
Expected *H. zea* moth catch as a function of (**a**) temperature, and (**b**) relative humidity in Jackson County, Florida. Solid lines indicate the fitted model (GLM with a negative binomial error distribution) and grey bands indicate the confidence intervals. Circles represent the total number of insects collected from commercial fields fortnightly, for two years, using sex-pheromone trapping (n = 339).

**Figure 6 insects-14-00395-f006:**
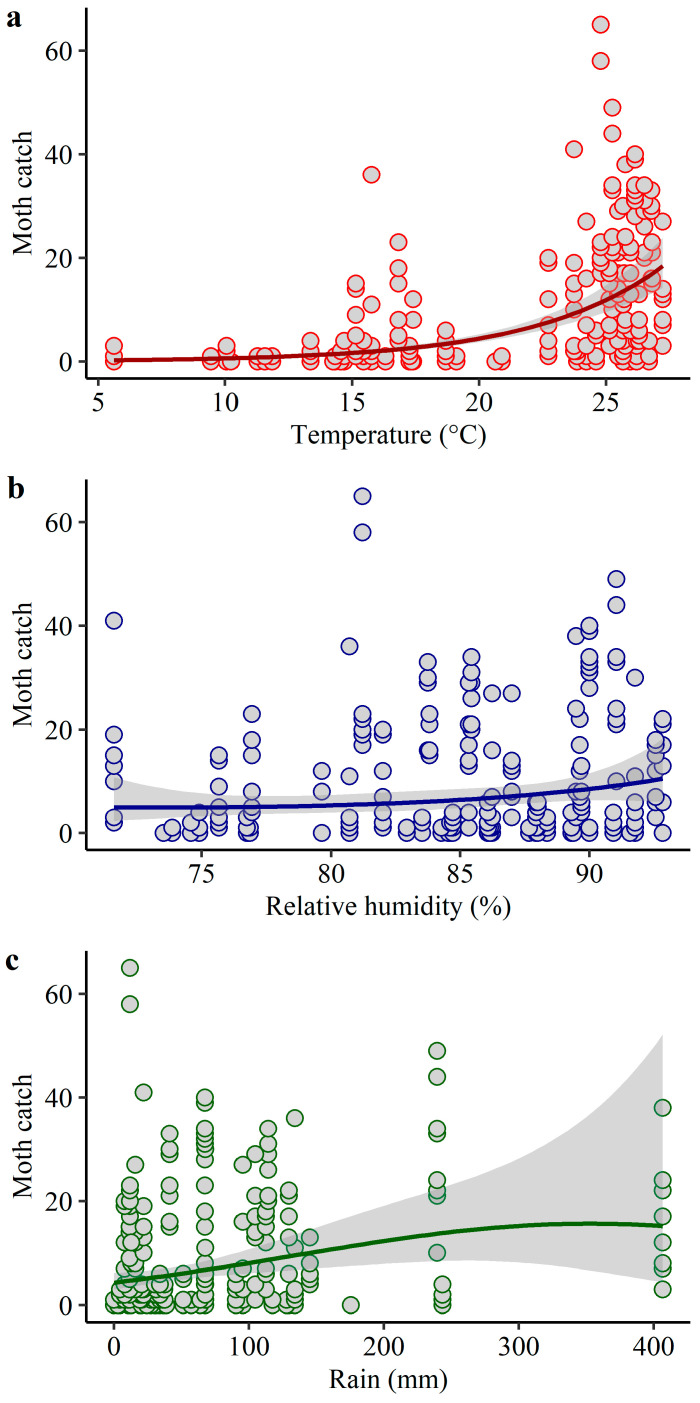
Expected *H. zea* moth catch as a function of (**a**) temperature, (**b**) relative humidity, and (**c**) rainfall in Santa Rosa/Escambia counties, Florida. Solid lines indicate the fitted model (GLM with a negative binomial error distribution) and grey bands indicate the confidence intervals. Circles represent the total number of insects collected for two years using sex-pheromone trapping (n = 335).

**Table 1 insects-14-00395-t001:** Wald χ^2^ type III analysis of variance for crop and irrigation system on the *H. zea* moth catch in Jackson and Santa Rosa/Escambia, Florida, from June 2017 to June 2019.

Region	Variable	χ^2^	DF	*p*-Value ^1^
Jackson	Crop	0.21	1	0.65
	Irrigation	0.15	1	0.90
	Crop × Irrigation	2.01	1	0.46
Santa Rosa/Escambia	Crop	0.54	1	0.46
	Irrigation	4.45	1	0.03
	Crop × Irrigation	0.13	1	0.72

^1^ *p*-values from generalized linear mixed model (negative binomial error distribution) lower than 0.05 were considered significant.

**Table 2 insects-14-00395-t002:** Weather variables association with *H. zea* moth catch in Jackson and Santa Rosa/Escambia, Florida, from June 2017 to June 2019.

Region	Parameter	Estimate	SD	*z* Value	Pr (>|z|)
Jackson	Intercept	−4.10	1.19	−3.46	0.0005
	Temperature	0.11	0.02	6.68	<0.0001
	Relative humidity	0.04	0.02	2.19	0.03
	Rainfall	−0.0002	0.00	−0.10	0.92
Nagelkerke’s R^2^					0.38
Escambia/Santa Rosa	Intercept	4.94	1.35	3.67	0.0002
	Temperature	0.23	0.02	14.16	<0.0001
	Relative humidity	−0.10	0.02	−5.76	<0.0001
	Rainfall	0.005	0.001	4.06	<0.0001
Nagelkerke’s R^2^					0.59

## Data Availability

The data presented in this study are available upon request to the corresponding author.

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
