# Peer review of "Factors Affecting Population Dynamics of *Helicoverpa zea* (Lepidoptera: Noctuidae) in a Mixed Landscape with Bt Cotton and Peanut"

_insects, 2023, doi:10.3390/insects14040395_

Round 1
Reviewer 1 Report
The authors report significant new information related to the population dynamics of H. zea on Bt and non-Bt crops across the Florida Panhandle area that have application to IRM and IPM in the region and surrounding areas. The paper reports year round pheromone sampling from multiple sites over two years, correlates this with weather and uses carbon isotope analysis to infer larval host plant dynamics over time.
Specific comments:
Line 140; You should better justify removing data sets collected on corn fields. Corn is a known host of H. zea and its presence may influence population dynamics in the area. Removing these fields may skew the results and not reflect agroecosystems in the area. Did you analyze the data with and without including those fields? If the results were similar in both cases, this could be stated.
Line 183; ‘was done’ should be ‘was analyzed’ or similar wording
Line 197; after ‘both’ insert ‘locations’
Line 239; after ‘14’ insert ‘h’
Line 339; ‘host crop hosts’ should be ‘crop hosts’
Several references are incomplete and should be edited to provide enough information for people to retrieve them;
Gassmann and Reisig (2023)
Reay-Jones (2019)
Westbrook and Lopez (2010)
Wolf (1979)
Several references have numbers inserted in the references that should be removed including:
Roach and Adkisson (1971), Stadelbacher (1981), Westbrook and Lopez (2010)
Author Response
We are submitting the revised version of the research manuscript " Helicoverpa zea (Lepidoptera: Noctuidae) in a mixed landscape with Bt and non-Bt crops: flight phenology, weather association, and host plant origin.” We appreciate the reviewer’s comments/suggestions and constructive criticisms. All the comments from the reviewer have been addressed in a detailed point-by-point response (below) and in the revised manuscript. All lines below refer to the clean version of the manuscript after all changes have been accepted. We believe we have an improved version now.
#Reviewer 01
Specific comments:
You should better justify removing data sets collected on corn fields. Corn is a known host of H. zea and its presence may influence population dynamics in the area. Removing these fields may skew the results and not reflect agroecosystems in the area. Did you analyze the data with and without including those fields? If the results were similar in both cases, this could be stated.
Similar results in the analysis with and without the corn data were detected in the preliminary analysis performed. However, as we have a limited number of fields cultivated with corn (one field in Jackson and three in Santa Rosa and Escambia), our results could be underestimating the role of corn on the population dynamics of H. zea in the agroecosystems. Due to this limitation on our dataset, we cannot draw conclusions regarding corn, and we removed those observations from the dataset. Based on the concern of the suggestion of the reviewer, we added the following information to the manuscript:
Lines 141- 144: “Preliminary analysis revealed no difference in H. zea flight phenology with or without data of H. zea moths trapped in corn fields. Due to the limited number of observations collected from corn fields in the agroecosystems under study, the corn trapped moths’ data were removed from the dataset in further analysis for consistency."
‘was done’ should be ‘was analyzed’ or similar wording Line 197; after ‘both’ insert ‘locations’
Both sentences were rewritten:
Lines 187-188: “… The association between weather variables and the abundance of H. zea moths caught was analyzed using a generalized linear model…”
Lines 202-203: “ In both locations, the highest H. zea abundance was in late summer…”
After ‘14’ insert ‘h’
Lines 246: The sentence was rewritten to “Photoperiod ranges from 10 to 14 h in both locations.”
‘host crop hosts’ should be ‘crop hosts’
Lines 341-342: The sentence was rewritten to “the contribution of crop hosts….”
Several references are incomplete and should be edited to provide enough information for people to retrieve them; Gassmann and Reisig (2023) Reay-Jones (2019) Westbrook and Lopez (2010) Wolf (1979).
The references were edited to provide complete information on each citation.
Lines 467-468: Gassmann, A.J.; Reisig, D.D. Management of Insect Pests with Bt Crops in the United States. Annu. Rev. Entomol. 2023, 68,31-49, doi:10.1146/annurev-ento-120220-105502.
Lines 461- 462: Reay-Jones, F.P.F. Pest Status and Management of Corn Earworm (Lepidoptera: Noctuidae) in Field Corn in the United States. J. Integr. Pest Manag. 2019, 10, 1-9, doi:10.1093/jipm/pmz017.
Lines 476 - 477: Westbrook, J.K.; López, J.D. Long-Distance Migration in Helicoverpa zea: What We Know and Need to Know. Southwest. Entomol. 2010, 35, 355–360, doi:10.3958/059.035.0315.
Lines 471-472: Wolf, W.W.; Westbrook, J.K.; Raulston, J.; Pair, S.D.; Hobbs, S.E. Recent Airborne Radar Observations of Migrant Pests in the United States. Philos. Trans. R. Soc. 1990, 328, 619–630, doi:10.1098/rstb.1990.0132
Several references have numbers inserted in the references that should be removed including Roach and Adkisson (1971), Stadelbacher (1981), Westbrook and Lopez (2010)
The numbers were removed from all references.
Lines 469-470: Roach, S.H.; Adkisson, P.L. Termination of Pupal Diapause in the Bollworm. J. Econ. Entomol. 1971, 64, 1057–1060, doi:10.1093/jee/64.5.1057.
Lines 431-432: Stadelbacher, E.A. Role of Early-Season Wild and Naturalized Host Plants in the Buildup of the F1 Generation of Heliothis zea and H. virescens in the Delta of Mississippi. Environ. Entomol. 1981, 10, 766–770, doi:10.1093/ee/10.5.766.
Lines 476 - 477: Westbrook, J.K.; López, J.D. Long-Distance Migration in Helicoverpa zea: What We Know and Need to Know. Southwest. Entomol. 2010, 35, 355–360, doi:10.3958/059.035.0315.
Reviewer 2 Report
The manuscript is interesting, but has major shortcomings.
C3, C4 decryption not entered in the text.
After mentioning the Stable carbon isotope (δ13C) analysis (Line 55), it would be appropriate to add C3 or C4 to the plant names (Lines 58-60). Give examples of wild plants which the first generation of H. zea uses as hosts (Line 58).
Lines 128-131. “The moths trapped on the collected sticky liners were identified using morphological characteristics of the black-colored spot on the forewings, a broad dark transverse band distally, and hind wings lighter in color (Hardwick 1965)”. – It is too unlikely to reveal morphological characteristics of sticky moths, especially those that have been collected by sticky liners for 2 weeks. In addition, many moths are eaten in traps by predatory insects, so an underestimation of the moth number is likely. Since the assessment of predation and its annual dynamics was not carried out, the number of trapped moths cannot fully reflect the number change of moths in nature. What is the level of specificity of pheromones? What insects besides Helicoverpa zea fell into the traps?
Line 133. Why did you use the word “adults”? Does this mean there is no certainty whether they were Helicoverpa zea males?
Line 142. An assembly of moths were - grammar mistake. Please specify whether this assembly of moths was from pheromone traps.
3.1. Annual phenology of H. zea flight. – It is necessary to add a comparative analysis of data by years and generations.
Line 207. Replace 3.1 with 3.2. – 3.2. Carbon isotopic signatures of H. zea moths. – It is necessary to add a comparative analysis of data by years and generations.
3.3. Weather in the Florida Panhandle and associations with moth catch. - It is necessary to add a comparison of the conditions of 2 identical periods of this two-year study.
Lines 272, 274. Give the most likely examples of plants C3 and C4 for every region and generation.
Line 479, 442, 447, 455, 502. Here, the sentences start with a lowercase letter.
Abbott et al., 2019 and Huffman et al., 1979 are not cited in the text, but they are in references (Lines 426, 481).
Author Response
We are submitting the revised version of the research manuscript " Helicoverpa zea (Lepidoptera: Noctuidae) in a mixed landscape with Bt and non-Bt crops: flight phenology, weather association, and host plant origin.” We appreciate the reviewer’s comments/suggestions and constructive criticisms. All the comments from the reviewer have been addressed in a detailed point-by-point response (below) and in the revised manuscript. All lines below refer to the clean version of the manuscript after all changes have been accepted. We believe we have an improved version now.
Suggestions for Authors
The manuscript is interesting but has major shortcomings.
C3, C4 decryption not entered in the text.
Lines 56 – 59: The decryptions of C3 and C4 plants were included, and it is now stated: “Stable carbon isotope (δ13C) analysis of moth wings has been used to investigate larval hosts of H. zea in the landscape among plants that produce a three- (C3) or a four- (C4) carbon compounds during the first step of the carbon fixation cycle in the photosynthesis pathway [13–15]. “
After mentioning the Stable carbon isotope (δ13C) analysis (Line 55), it would be appropriate to add C3 or C4 to the plant names (Lines 58-60). Give examples of wild plants which the first generation of H. zea uses as hosts (Line 58).
Lines 59- 64: The names of C3 and C4 plants and examples of wild hosts were added “Generally, in the southeastern United States, the first generation of H. zea mainly uses C3 wild plants as hosts (such as crimson clover (Trifolium incarnatum) [4], pepperweed (Lepidium sp.)[3], and wild toadflax (Nuttallanthus texanus)[1,5]. The second generation feeds on corn (Zea mays L.), a C4 plant, and subsequent generations switch to late-season C3 crops such as cotton (Gossypium hirsutum L.), soybean (Glycine max L.), and peanut (Arachis hypogaea L.) [14].”
Lines 128-131. “The moths trapped on the collected sticky liners were identified using morphological characteristics of the black colored spot on the forewings, a broad dark transverse band distally, and hind wings lighter in color (Hardwick 1965)”. – It is too unlikely to reveal morphological characteristics of sticky moths, especially those that have been collected by sticky liners for 2 weeks. In addition, many moths are eaten in traps by predatory insects, so an underestimation of the moth number is likely. Since the assessment of predation and its annual dynamics was not carried out, the number of trapped moths cannot fully reflect the number change of moths in nature.
What is the level of specificity of pheromones? What insects besides Helicoverpa zea fell into the traps?
Pheromone traps are a common technique used for studying the seasonal fluctuation of noctuid insects. There are different designs of pheromone traps, and for this study, we used delta traps with sticky liners. The sticky liners are placed inside the delta traps and provide protection against damage on the moths that could jeopardize species identification, based on morphological characteristics. Rainfall was our biggest concern for the region under study, and preliminary analysis showed better integrity of morphological characteristics of moths collected from delta traps than bucket traps. Besides, delta traps require a lower amount of time for maintenance, and it is easier to store sticky liners under the freezer for late evaluation. Regarding predation, the glue in the sticky liners impedes non-target or predators, such as coleopterans, from moving and destroying specimens. In an episode, a bird was stuck in the sticky liners, but the H. zea moths were not destroyed and recovered.
Cross-attraction throughout the H. zea pheromone trapping study was observed. The cross attracted species were Spodoptera frugiperda, Elasmopalpus lignosellus, Mythimna unipuncta, and Leucania sp. We did not record the densities of these other species, and the remarkable morphological differences of those species allowed us to distinguish them from H. zea moths, following morphological characteristics describes on lines 129-132: “Trapped H. zea moths were identified using morphological characteristics of the black-colored spot on the forewings, a broad dark transverse band distally, and hind wings lighter in color [6]”.
Why did you use the word “adults”? Does this mean there is no certainty whether they were Helicoverpa zea males?
Lines 134 -135: Adults refer to Helicoverpa zea moths. The sentence was rewritten for clarity, as it reads state “… the total number of H. zea moths collected in both traps was determined.”
An assembly of moths were - grammar mistake. Please specify whether this assembly of moths was from pheromone traps.
Lines 147-148: The sentence was rewritten for clarity, as it now stated: “A sample of trapped H. zea of moths were selected and submitted to carbon isotopic analysis to identify larval feeding on C3 or C4 host plants.”
3.1. Annual phenology of H. zea flight. – It is necessary to add a comparative analysis of data by years and generations.
In this study, we documented the monthly fluctuation of H. zea moth in the landscape of the Florida Panhandle. The information on the population dynamic throughout the year contributes to the improvement of the IPM and IRM of this economic pest for the region, especially related to the phenology of the occurrence of the pest. The year was used as a replication to better estimate moth abundance in each month. The work did not focus on comparing changes in H. zea population densities through the years. Further studies focusing on comparing differences among years, for example, studying the trend in H. zea population over time or considering the ENSO cycle, would be interesting. However, the authors believe that documentation of temporal changes in the densities of this species would demand long-term data collection.
Regarding generations, no study has determined the number of H. zea generations in the Florida Panhandle. During this study, H. zea moths were trapped in all months of the year. Helicoverpa zea moths trapped during early spring, for example, could potentially originate from a local population emerging from diapausing pupae from the soil or a migratory population from other regions, such as south Florida. Thus, analysis by generation is not possible.
Line 207. Replace 3.1 with 3.2. – 3.2. Carbon isotopic signatures of H. zea moths. – It is necessary to add a comparative analysis of data by years and generations.
For this study, we aimed to investigate larval host plants for the trapped H. zea moths over the seasons. We are not focused on comparing the change in host usage over the years, especially because the crop cultivated during these two years were stable. Besides, a low number of H. zea moths were trapped during the winter months. Combining H. zea adults for both years allowed us to increase the number of moths evaluated per season.
As discussed in the previous answer, comparison among generations would not be possible.
3.3. Weather in the Florida Panhandle and associations with moth catch. - It is necessary to add a comparison of the conditions of 2 identical periods of this two-year study.
Since the study did not focus on detecting differences among years, a comparison of the weather conditions among two identical periods was not included. The authors agree with the importance of weather parameters in the H. zea population dynamics. In section 3.3 we provided a complete description of the weather description (and oscillation) for each study location (lines 238-249). In addition, the potential effect of the weather on H. zea population dynamics and development was discussed on lines 357-394.
Give the most likely examples of plants C3 and C4 for every region and generation.
Lines 282 – 284: In this first paragraph of the discussion, we summarized the overall results from our study. In the following paragraphs, the most likely C3 and C4 plants are discussed, based on the prevalent crops cultivated in the Florida Panhandle agroecosystem. In lines 303 – 309 the sentence was updated to: “Cotton and peanut are the main C3 crops cultivated in Florida Panhandle, but low acreages of soybeans and tomatoes are also observed. Considering the cultivation of Bt cotton in the region, peanut could therefore support large populations of H. zea larvae, playing a role as a source of moths with C3 larval hosts in the Florida Panhandle. Due to H. zea flight capacity , moths trapped in cotton fields could have originated from larvae that fed on peanut or other cultivated and non-cultivated C3 hosts.” In lines 313 – 315 “Although corn is less than 5% of the total acreage cultivated in the region, this crop can be considered the prevalent plant with C4 photosynthetic pathway in both the landscapes during the crop season.”
Line 479, 442, 447, 455, 502. Here, the sentences start with a lowercase letter.
Answer: The corrections have been included.
Abbott et al., 2019 and Huffman et al., 1979 are not cited in the text, but they are in references (Lines 426, 481).
Answer: Both references were removed, and bibliography was checked.
Round 2
Reviewer 2 Report
I carefully read the new version of the manuscript and author's notes. First of all, it should be noted that the authors have corrected all minor comments. I am pleased with the understanding of the authors that “documentation of temporal changes in the densities of this species would demand long-term data collection”. I believe that if misleading and speculative reasoning is removed from the manuscript, and the methodology, results and conclusions of this study are described in more detail and more specifically, then the article can be published in some regional journal, but not Insects.
The title states “Helicoverpa zea (Lepidoptera: Noctuidae) in a mixed landscape with Bt and non-Bt crops”, however, the manuscript does not indicate the percentage of areas of Bt and non-Bt crops in the studied regions and nearby, from where the pest can migrate. Flight phenology was not studied in this study. For this, other methods must be used. “Нost plant origin” is not relevant to this study at all. Therefore, starting with the title, the authors mislead the reader.
Simple summary is overloaded with information more suitable for an introduction: “Due to its polyphagous nature, H. zea is known by several common names, including corn earworm, tomato fruitworm, and cotton bollworm. Weather, host distribution, and abundance influence H. zea population dynamics. In North America, H. zea overwinters in southern states and recolonizes areas in northern states, which have lethal winter temperatures for this species. Studies of H. zea ecology in southern states, such as Florida, that are sources of migrants can provide region-specific information with potential continental-wide implications”.
The study was conducted from 2017 to 2019, so it is not clear why Materials and Methods refer to 2020.
Figure 4 does not make sense, as it does not show a relationship with any indicators of insect abundance.
I consider it inappropriate to further consider the shortcomings of the article, since it should not be published in this journal due to an unsuitable methodology for studying phenology, the lack of scientific novelty in the study of the correlation of weather conditions with the flight of moths.
Round 3
Reviewer 2 Report
Please compare final sentences of Simple Summary and Abstract. In my opinion, these phrases “larvae feeding predominantly on C3 host plants, with greater larval feeding on C4 hosts occurring during the summer” and “C3 plants were the main H. zea hosts throughout the year, although some feeding on C4 hosts was likely in the summer months” have different meanings.
During the period of mass flight, as well as in the hot season, traps should be checked not once in 2 weeks, but at least 4 times (Sharp et al., 1978). Sharp JL, McLaughlin JR, James J, Eichlin TD, Tumlinson JH. Seasonal occurrence of male Sesiidae in north central Florida determined with pheromone trapping methods. Florida Entomologist. 1978 Dec 1:245-250.
It is interesting to know the size of the sticky liner and the maximum number of moths in 2 weeks. Sometimes during flight peak, moths able to cover the entire surface of the sticky liner in 1 day.
The methodology should describe the crops (Bt or non-Bt) grown in the fields, where the traps were installed, a map-scheme of the location of the fields and the placement of the traps.
On which crops (peanut or cotton, Bt or non-Bt)/fields (irrigated or rainfed) were more moths caught, on which less? Were the differences significant? Have any patterns been identified?
Lines 71-73. The temporal separation of corn and cotton can create a sink-to-source relationship of H. zea feeding following the availability and attractiveness of these crops, potentially selecting genes that confer resistance to Bt toxins [13,14,16]. – Perhaps, this sentence as well as the term “mixed Bt and non-Bt crops” would have ensured a better fit to the scope of the special issue if that had been the purpose of the study and the corresponding results had been obtained.
The manuscript as it is currently submitted have to be rejected.
Author Response
Comments and Suggestions for Authors
Please compare final sentences of Simple Summary and Abstract. In my opinion, these phrases “larvae feeding predominantly on C3 host plants, with greater larval feeding on C4 hosts occurring during the summer” and “C3 plants were the main H. zea hosts throughout the year, although some feeding on C4 hosts was likely in the summer months” have different meanings.
Answer: Both sentences were rewritten. In the simple summary, lines 28-29, now read: “Carbon isotopic analysis indicated H. zea larval feeding predominantly on C3 host plants throughout the year. However, during the summer, larval feeding on C4 hosts was detected. On the abstract lines 42-43 “C3 plants were the main H. zea hosts throughout the year, but larval feeding on C4 hosts was detected during summer months.”
During the period of mass flight, as well as in the hot season, traps should be checked not once in 2 weeks, but at least 4 times (Sharp et al., 1978). Sharp JL, McLaughlin JR, James J, Eichlin TD, Tumlinson JH. Seasonal occurrence of male Sesiidae in north central Florida determined with pheromone trapping methods. Florida Entomologist. 1978 Dec 1:245-250.
Answer: Preliminary trapping was conducted in the region when designing the present study to determine the proper interval to check the traps. This initial study was performed during the summer when we expected high densities of H. zea moths, and we detected the two-week interval as appropriate for checking the liners. We did not observe liners completely covered by moths in any situation during the two years of the study. Besides, the publications cited indicated data collected for species of the family Sesiidae prior to the cultivation of transgenic Bt technology. Densities of Noctuidae species that are targeted by Bt technology, such as H. zea has been decreased over time due to area-wide suppression (Adamczy and Hubbard, 2006), and this could be one reason for the population densities detected in this study being relatively lower than prior Bt technology.
Adamczyk, J.J.; Hubbard, D. Changes in Populations of Heliothis virescens (F.) (Lepidoptera: Noctuidae) and Helicoverpa zea (Boddie) (Lepidoptera: Noctuidae) in the Mississippi Delta from 1986 to 2005 as Indicated by Adult Male Pheromone Traps. J.Cotton Sci.2006, 10,155-160..
It is interesting to know the size of the sticky liner and the maximum number of moths in 2 weeks. Sometimes during flight peak, moths are able to cover the entire surface of the sticky liner in 1 day.
Answer: The authors share the same concern as the reviewer. As previously indicated, we did not detect the problem of moths completely covering the liners. The sticky liners are 18.3 cm x 18.3 cm. The maximum number of moths caught in a two-week interval was 25 moths in Jackson, and 38 moths in Santa Rosa/ Escambia counties. In both situations, the specimens were well preserved, and we are able to identify the species and recover them.
The methodology should describe the crops (Bt or non-Bt) grown in the fields, where the traps were installed, a map scheme of the location of the fields and the placement of the traps.
Answer: The section of the methodology that describes the crops was rewritten for clarity. In lines 107 to 111, it is stated: “Four commercial fields averaging 25 hectares of cotton and four fields of peanut were selected in each region, and two delta Pherocon® VI Traps (Trécé Incorporated) were mounted on the edges of each field on aluminum poles, approximately one meter above the ground to collect H. zea moths. In each region, four of the selected fields (two cotton and two peanut fields) were under irrigation and the other four fields were rainfed”. In addition, in Table 1, a column with information on the crop cultivated (peanut or Bt cotton) in each field was included. In addition, the information that 100% of the cotton cultivated in the region is transgenic Bt cotton is presented in lines 84 to 86.
On which crops (peanut or cotton, Bt or non-Bt)/fields (irrigated or rainfed) were more moths caught, on which less? Were the differences significant? Have any patterns been identified?
Answer: The effect of crop and irrigation were tested, and we found no effect of the crop, but a significant effect of the irrigation system in the Santa Rosa/ Escambia region. We have added this information to the manuscript and in the material and methods, lines 170 to 173 we included the following sentence: “On a third model we tested the effect of crop (peanut or cotton) and irrigation system (irrigated or rainfed) in H. zea moth trap catches for each region using a GLMM. Crop and irrigation systems were set as fixed effects and farm and month were set as random effects.” In the results section, in lines 205-208, we have included the following sentence: “No effect for crop, irrigation system, or crop x irrigation system interaction was observed for trap catches in Jackson County (Table 02). However, a significant effect was observed for the irrigation system (χ2 =4.45, P=0.03) in Santa Rosa/Escambia (Table 02), with higher catches in rainfed fields”. Besides, a table was included in the results section presenting the analysis of crop and irrigation systems on the H. zea catch (Table 02). For the discussion section, in lines 310 to 321, we added “No differences in H. zea moth catches were detected between traps set on cotton versus peanut crops. Considering the cultivation of Bt cotton in the region and the efficacy of this technology on suppressing H. zea populations [36] peanut is expected to support large populations of H. zea larvae, playing a role as a source of moths with C3 larval hosts in the Florida Panhandle. Due to H. zea flight capacity [23,24], moths trapped in cotton fields could be originated from larvae that fed on peanut or other cultivated and non-cultivated C3 hosts. Noteworthy, during the peak of H. zea abundance detected in this study, cotton is on the reproductive stage, which is highly attractive for H. zea. The high acreages of Bt cotton cultivation can lead to a high risk for resistance selection. Thus, cotton fields in Florida Panhandle should be monitored as a proactive approach to detect early escapes of this Bt technology, and to determine the role of cotton as a source of C3 feed moths.”
In addition, we added the following sentence in the discussion section on lines 325-332 “A significant effect was observed for irrigation system in Santa Rosa/ Escambia region, with higher catches in rainfed fields. Accordingly, higher H. zea larval densities have been detected in rainfed soybeans fields [48], and irrigation has been shown to cause H. zea pupal mortality in cotton fields [49]. The impact of irrigation on the immature stages of H. zea could explain the effect of irrigation in Santa Rosa/ Escambia trap catches. However, this study focused on determining the monthly abundance of H. zea moths, and we cannot draw conclusions about factors regulating H. zea immature survival in rainfed and irrigated fields. Future studies should investigate the effects of irrigation on H. zea population dynamics in Florida agroecosystem.”
Lines 71-73. The temporal separation of corn and cotton can create a sink-to-source relationship of H. zea feeding following the availability and attractiveness of these crops, potentially selecting genes that confer resistance to Bt toxins [13,14,16]. – Perhaps, this sentence as well as the term “mixed Bt and non-Bt crops” would have ensured a better fit to the scope of the special issue if that had been the purpose of the study and the corresponding results had been obtained.
Answer: We agree with the reviewer's comment. However, this study had a focus on documenting H. zea moth abundance in a mixed landscape, with the predominant crops being Bt cotton and peanut, with smaller acreage of Bt corn cultivated during the beginning of the summer. The carbon analysis was used to determine feeding on C4 (especially corn) and C3 hosts (such as peanut, and cotton) and to investigate a potential succession of feeding on this two Bt technology in the landscape. Because cotton and peanut are the predominant crops in the region under study, the mixed bt and non-bt crops refer primarily to these two crops. We also call the attention that no differences were detected on trap catches among these two crops. Due to the high efficacy of Bt cotton technology in the area under study and H. zea flight capacity, we hypothesized that the H. zea moths caught in Bt cotton are probably originated from peanut, or other C3 hosts, as discussed in lines 311 -314.
The manuscript as it is currently submitted have to be rejected.
Answer: An updated version has been submitted, which visited all the comments and suggestions. We believe we have improved the manuscript.